# An Entropy-Based Neighborhood Rough Set and PSO-SVRM Model for Fatigue Life Prediction of Titanium Alloy Welded Joints

**DOI:** 10.3390/e21020117

**Published:** 2019-01-27

**Authors:** Li Zou, Yibo Sun, Xinhua Yang

**Affiliations:** 1Software Technology Institute, Dalian Jiaotong University, Dalian 116028, China; 2Dalian Key Laboratory of Welded Structures and Its Intelligent Manufacturing Technology (IMT) of Rail Transportation Equipment, Dalian Jiaotong University, Dalian 116028, China; 3Sichuan Provincial Key Lab of Process Equipment and Control, Zigong 643000, China

**Keywords:** fatigue life prediction, welded joints, neighborhood entropy, PSO, SVRM

## Abstract

In order to obtain comprehensive assessment of the factors influencing fatigue life and to further improve the accuracy of fatigue life prediction of welded joints, soft computing methods, including entropy-based neighborhood rough set reduction algorithm, the particle swarm optimization (PSO) algorithm and support vector regression machine (SVRM) are combined to construct a fatigue life prediction model of titanium alloy welded joints. By using an entropy-based neighborhood rough set reduction algorithm, the influencing factors of the fatigue life of titanium alloy welded joints such as joint type, plate thickness, etc. are analyzed and the reduction results are obtained. Fatigue characteristic domains are proposed and determined subsequently according to the reduction results. The PSO-SVRM model for fatigue life prediction of titanium alloy welded joints is established in the suggested fatigue characteristic domains. Experimental results show that by taking into account the impact of joint type, the PSO-SVRM model could better predict the fatigue life of titanium alloy welded joints. The PSO-SVRM model indicates the relationship between fatigue life and fatigue life influencing factors in multidimensional space compared with the conventional least-square S-N curve fitting method, it could predict the fatigue life of the titanium alloy welded joints more accurately thus helps to the reliability design of the structure.

## 1. Introduction

Welded components have been widely used in the engineering equipment field under static or dynamic load circumstances because of their advantages of good connectivity, light weight and ease of use. Due to stress concentration however, the welded joint becomes one of the weak links of the structural reliability of the product. This causes great hidden dangers to the safety and normal work of the structure.

Currently, three popular fatigue life analysis methods of welded joints have been proposed, which include S-N curve method, local stress strain method and fracture mechanics method [1]. Compared with the local stress strain method and fracture mechanics method, the S-N curve method is more widely used in the design stage which mainly consists of nominal stress method [2], hot spot stress method [3,4] and nodal force-based structural stress method [5,6]. Due to the mesh insensitivity of structural stress calculations, the higher accuracy of fatigue life prediction and the wide applicability, the nodal force-based structural stress method is one of the most notable fatigue life evaluation methods for welded structures. Dong reprocessed thousands of fatigue test data of steel welded joints from the last 50 years [7]. The research results indicated that when taking into account the influences of joint thickness and load model, the S-N sample data is compressed into a narrow band. In the present study, comprehensive and quantitative analysis of the influencing factors which affect the fatigue life of welded joints is lacking and the problem of low precision of fatigue life prediction of welded joints still exists.

Thermodynamics entropy has been used for fatigue life prediction for a long time. For example, a damage metric based on the second law of thermodynamics and statistical mechanics was presented [8]. The proposed thermodynamic framework treated a solid body as a thermodynamic system and required that the entropy production be non-negative. A thermodynamic framework was presented for damage mechanics of solid materials [9]. It was believed that there was no need for physically meaningless empirical parameters to define a phenomenological damage potential surface or a Weibull function to trace damage evolution in solid continuum. A generalized theory of evolution based on the concept of tribo-fatigue entropy was proposed where tribo-fatigue entropy was determined by the processes of damage ability conditioned by thermodynamic and mechanical effects causing to the change of states of any systems [10]. The Shannon’s differential entropies of both distributions of the strain in the loading direction and the first principal strain were employed at the tension peak of the cycles as measuring parameters of strain inhomogeneity [11]. A thermo- mechanical fatigue life prediction model based on the theory of damage mechanics was presented in [12]. The damage evolution, corresponding to the material degradation under cyclic thermo mechanical loading, was quantified thermodynamic framework. A viscoplastic constitutive model unified with a thermodynamics based damage evolution model was embedded into a couple stress framework in order to simulate low cycle fatigue response coupled to size effects [13]. Entropy, as a state function of thermodynamics, is independent of the failure path and can deal with the uncertainties related to the damage threshold [14] (such as the systemic chaos caused by performance degradation). Entropy generation and entropy accumulation are taken as the measurement of performance degradation [15], and the effectiveness of using constant irreversible entropy increment to evaluate metal fatigue damage [14] is also gradually verified.

Meanwhile, from the perspective of information entropy, the principal author’s team has proposed an evaluation model of aluminum alloy welded joint low-cycle fatigue data based on information entropy [16]. In order to obtain the comprehensive assessment of fatigue life influencing factors of the titanium alloy welded joints, to further reduce the dispersion level of S-N data samples and to improve the accuracy of fatigue life prediction of the welded joints, the concept of fatigue characteristics domain is proposed by using entropy based neighborhood rough set theory and the PSO-SVRM model for fatigue life prediction of welded joints is established in this work.

## 2. Nodal Force Based Structural Stress Principle

In the finite element stress calculation the results are affected by mesh size, which causes inconsistency in calculations for different structures. In order to address this problem, structural stress based on the line force is first defined in the nodal force based structural stress method.

The normal structural stress at each node is defined as
(1)$$\sigma_{s}=\sigma_{m}+\sigma_{b}$$
(2)$$\sigma_{m}=\frac{Fy}{A}=\frac{Fy}{l\cdot t}$$
(3)$$\sigma_{b}=\frac{M_{x}}{W}=\frac{M_{x}}{\frac{1}{6}\cdot l\cdot t^{2}}$$
where *F_y_* is the vertical stress in the weld toe, *M_x_* is the moment around the weld toe. $f_{y}=Fy/l$ is the line force and $m_{x}=M_{x}/l$ is the line moment as shown in Figure 1. Structural stress can be expressed as [5]:(4)$$\sigma_{s}=\sigma_{m}+\sigma_{b}=\frac{f_{y}}{t}+\frac{6m_{x}}{t^{2}}$$

An equivalent structural stress range parameter can be defined as:(5)$$\Delta S_{s}=\frac{\Delta \sigma_{s}}{t^{\frac{2-m}{2m}}\cdot I{\left(r\right)}^{\frac{1}{m}}}$$
where $\Delta \sigma_{S}$ represents the structural stress range calculated, I(r) is a dimensionless parameter derived by fracture mechanics considerations, and *m* is the crack propagation exponent in the conventional Paris law, taking on a value of about 3.6 [17]. It could be seen that the equivalent structural stress parameter described in Equation (5) can capture the effects of stress concentration, plate thickness and loading mode effects on fatigue behavior of welded components.

The formula for fatigue life calculation of welded joints using the equivalent structural stress $\Delta S_{s}$ can be expressed as:(6)$$\Delta S_{s}=CN^{{}_{h}}$$
where *N* is the number of cycles which indicate the fatigue life of the structure, *C* is material constant and *h* represents the negative slope of the master S-N curve.

## 3. Entropy Based Neighborhood Rough Set Model

Basic concepts are introduced in this section before the entropy-based neighborhood rough set model is established. Fatigue life influencing factors of the titanium alloy welded joints are analyzed by using the established model. Key influencing factors together with the weights are obtained and the fatigue characteristic domains are subsequently determined. 

### 3.1. Basic Concept

Proposed by Pawlak, rough set theory [18] has been used successfully as a new feature reduction tool to discover data dependencies and reduce the number of features, but the classical rough set can only operate effectively with data sets containing categorical values. When using classical rough set theory, discretion must be done beforehand which inevitably result in loss of information. To overcome this drawback, Lin [19] proposed a neighborhood rough set model by extending the equivalence relation of classical rough sets. The neighborhood rough set model can process both numerical and categorical data set efficiently [20].

Uncertainty measure is an important way to describe classification ability and classification accuracy in rough set theory. Many scholars have studied this. For example, Chen proposed the concept of neighborhood entropy [21] and suggested several uncertainty measures including neighborhood accuracy, information quantity, neighborhood entropy and information granularity in the neighborhood systems [22]. Uncertain measures like roughness measure, accuracy measure, entropy and granularity for covering rough set models were studied and a new kind of partial order was suggested [23]. Huang [24] discussed the uncertainty measures of information entropy and rough entropy.

Feature reduction is a quite useful data preprocessing technique, aiming to obtain a minimal feature subset from a decision system while maintaining the same classification accuracy by deleting the noisy, irrelevant or misleading features. An uncertainty measure method of neighborhood entropy to evaluate the quality of the reduction results is proposed for neighborhood rough set reduction algorithm in this work. The fatigue characteristic domain is then determined on basis of the reduction results for classification of the fatigue samples to further reduce the dispersion level.

Several definitions about the basic concept of neighborhood rough set theory would be introduced first before further discussion of the neighborhood entropy based reduction algorithm.

**Definition 1.** *A neighborhood decision system can be denoted as*ND=(U,C∪D,δ)*and a distance function*Δ(x,y)→[0,1]. *Where U = {x_1_, x_2_, …, x_n_} is a nonempty finite set of objects called the universe, C = {a_1_, a_2_, …, a_m_} is the condition feature set, D is the set of decision features, and*δ*is the neighborhood parameter (*0≤δ≤1*).*Δ(x,y)*is a distance function, which satisfies:**(1)* Δ(x,y)≥0;*(2)* Δ(x,y)=0*, if and only if**x* = *y*;*(3)* Δ(x,y)=Δ(y,x);*(4)* Δ(x,y)+Δ(y,z)≥Δ(x,z).

For a feature subset B⊆C and a neighborhood parameter δ∈[0,1], a similarity relation denoted by:(7)$$NR_{\delta }(B)=\{(x,y)\in U\times U|\Delta (x,y)\leq \delta \}$$

The neighborhood class $n_{B}^{\delta }(x)$ of *x* ∈ *U* in the subspace *B* is defined as:(8)$$n_{B}^{\delta }(x)=\{y|x,y\in U,f(x,y)\leq \delta \}$$

**Definition 2.** 
*Let*
ND=(U,C∪D,δ)
*be a neighborhood decision system,*
B⊆C
*be a feature subset, for any object set*
X⊆U
*, the neighborhood lower and upper approximations of X on B are defined as:*
(9)$$\underline{N_{B}}(X)=\{xi\in U|n_{B}^{\delta }(x)\subseteq X\}$$
(10)$$\bar{N_{B}}(X)=\{xi\in U|n_{B}^{\delta }(x)\cap X\neq \phi \}$$


**Definition 3.** 
*Let*
ND=(U,C∪D,δ)
*be a neighborhood decision system,*
B,M,N⊆C
*be the feature subsets, the neighborhood entropy of B is denoted by:*
(11)$$E_{\delta }(B)=-\frac{1}{\left|U\right|}\sum_{i=1}^{\left|U\right|}\log_{2}\frac{1}{\left|n_{B}^{\delta }(x)\right|}$$
*and the neighborhood joint entropy of M and N is denoted by:*
(12)$$E_{\delta }(M,N)=-\frac{1}{\left|U\right|}\sum_{i=1}^{\left|U\right|}\log_{2}\frac{1}{\left|n_{M}^{\delta }(xi)\cap n_{N}^{\delta }(xi)\right|}$$


Let U/D={D1,D2,…,Dn} be equivalence classes constituted by decision feature set *D* on the universe *U*, the neighborhood conditional entropy of *D* on *B* is denoted by:(13)$$E_{\delta }(D|B)=-\frac{1}{\left|U\right|}\sum_{i=1}^{\left|U\right|}\log_{2}\frac{\left|n_{B}^{\delta }(xi)\right|}{\left|n_{B}^{\delta }(xi)\cap n_{D}^{\delta }(xi)\right|}$$

**Definition 4.** 
*Let*
ND=(U,C∪D,δ)
*be a neighborhood decision system,*
B⊆C
*be a feature subset,*
∀a∈C−B
*the significance of condition attribute a is defined as:*
(14)$$SIG(a,B,D)=E_{\delta }(D|B\cup a)-E_{\delta }(D|B)$$


**Definition 5.** 
*Let*
ND=(U,C∪D,δ)
*be a neighborhood decision system,*
B⊆C
*be a feature subset, we say B is a reduction of C if:*
(15)$$E_{\delta }(D|C)=E_{\delta }(D|B)$$
(16)$$\mathrm{and}\;\forall a\in B,E_{\delta }(D|B)>E_{\delta }(D|B-a)$$


### 3.2. Entropy-Based Neighborhood Reduction Algorithm

An uncertain measurement of neighborhood entropy-based reduction algorithm is suggested in this work. Generally speaking, there are five main steps in the algorithm. The flow chart of the entropy-based neighborhood reduction algorithm is shown as Figure 2 and the detailed steps of the algorithm are shown in [Sec secA1-entropy-21-00117].

In the figure *ND* denotes that the neighborhood decision system is the input of the algorithm and red indicating the reduction result set is the output of the algorithm. The initial *red* is set to be ϕ at the beginning. In each iteration of the algorithm, significance of each condition attribute is calculated according to Equation (14), then the condition attribute *a_k_* which has the max significance is selected. Determine whether the significance of the selected attribute *a_k_* is bigger than 0. If it is bigger than 0, *a_k_* is added to the reduction result sets *red*, the next iteration of the algorithm will begin, otherwise, the algorithm stops and output the reduction results *red*.

### 3.3. Fatigue Characteristic Domain Determination

Commercial pure titanium has already been used for seawater cooling pipes in thermal power plants because of its high corrosion resistance in seawater. Additionally, the advantage of its high specific strength has already been proved on racing yachts, although there is still a lack of research concerning the fatigue strength of titanium fillet welded joints that are indispensable for ship structures.

In this work, a neighborhood decision system (Table 1) of titanium alloy welded joints was established based on the well-documented fatigue data. Forty three (43) sets of titanium alloy fatigue data [25] were cited. The test specimens were manufactured using 2 and 10 mm thick commercial pure titanium mill products (JISH4600 TP340C/H). They were made with their lengths both along and transverse to the plate rolling direction, referred to as L or C, respectively. Furthermore, the butt-welded joints, transverse fillet welded joints or longitudinal fillet welded joints are referred to as B, T or L, respectively. The five-types of welded specimen (LB, CB, LT, CT and LL) were instrumented with strain gauges to determine the nominal stress distribution approaching the weld in the tensile test. Equivalent structural stress range was then calculated according to nodal force- based structural stress principle detailed in Section 2. As could be seen from Table 1, there were five condition features including joint type (*J*), plate thickness (*t*), load ratio (*r*), material name (*M*) and equivalent structural stress range ($\Delta S_{s}$) and one decision feature {life cycles (*N*)} in the decision system.

Entropy-based neighborhood rough set reduction algorithm was used here for attributes reduction as described in Section 3.2. The attribute reduction result of the decision system of titanium alloy welded joints was {*J*,$\Delta S_{s}$}. As could be seen from the reduction result, joint type (*J*) is one of the most important influencing factors of the titanium alloy welded joints besides the equivalent structural stress range ($\Delta S_{s}$). This indicates that the joint type influencing factor could not be omitted, or in other words, the joint type should be considered for fatigue life prediction of the titanium alloy welded joints in the nodal force based structural stress method. Fatigue characteristic domain of joint type is thus proposed. All the fatigue test samples are then divided into 5 different domains (S_1_–S_5_ in Figure 3) according to different joint type LB, CB, LT, CT and LL.

As can be seen from Figure 3, an “automatic” classification of fatigue specimen samples could be achieved when the fatigue characteristic domains are determined. This classification is based on the theory of knowledge granularity. The most important fatigue life influencing factors of the titanium alloy welded joints are obtained through attribute reduction by using entropy based neighborhood reduction algorithm.

## 4. PSO-SVRM Model for Fatigue Life Prediction of Titanium Alloy Welded Joints

### 4.1. PSO Algorithm 

PSO algorithm is an evolutionary computing technology proposed by Eberhart and Kennedy in 1995 [26]. It originates from the study of bird prey behaviour. Similar to other evolutionary algorithms, PSO is a global search algorithm based on iterative updating. In PSO, multiple candidate solutions coexist and cooperate with each other simultaneously. It searches for the optimal solution through cooperation and information sharing among individuals in a group, and the convergence speed of the algorithm is fast. It has been widely used in function optimization, pattern recognition and other fields nowadays.

Suppose in a D-dimension target search space, the particle swarm consist of *m* particles, the position of the *i*-th (*i* = *1*, *2*, …, *m*) particle is marked as $X_{i}^{t}=(X_{i1}^{t},X_{i2}^{t},\ldots ,X_{iD}^{t})$ and the velocity is marked as $V_{i}^{t}=(V_{i1}^{t},V_{i2}^{t},\ldots ,V_{iD}^{t})$. The optimal personal position that the *i*-th particle has experienced is marked as $P_{i}^{t}=(P_{i1}^{t},P_{i2}^{t},\ldots ,P_{iD}^{t})$ while the optimal position that the whole particle swarm has experienced is marked as $P_{g}^{t}=(P_{g1}^{t},P_{g2}^{t},\ldots ,P_{gD}^{t})$.The position and velocity of each particle in the swarm are updated by Formula (16) [27,28]:(17)$$\left\{ \begin{array}{l} V_{id}^{t+1}=\omega^{t}\cdot V_{id}^{t}+c_{1}\cdot rand\cdot (P_{id}^{t}-X_{id}^{t})+c_{2}\cdot rand\cdot (P_{gd}^{t}-X_{id}^{t})\\ X_{id}^{t+1}=X_{id}^{t}+V_{id}^{t+1},d=1,2,\ldots ,D \end{array}\right.$$
where *i* = 1, 2, …, *m*, $\omega^{t}=\omega_{\max}-\frac{\omega_{\max}-\omega_{\min}}{t_{\max}}\times t$ is an inertia weight that controls the diversification feature of the algorithm, *c*_1_ and *c*_2_ are learning factors that control the intensification feature of the algorithm, *rand* is a random number between (0,1).

The velocity of each particle *V_id_* in each dimension is usually limited between [−*V_ma_*_x_, *V_max_*], in order to prevent the particle from being too fast to miss the optimal solution. The algorithm ends when the maximum number of iterations is achieved or the specified accuracy requirements are met.

### 4.2. SVRM Principle

SVRM [29] is a supervised machine learning algorithm based on statistical learning theory with finite samples. In the nonlinear case, the input vector is mapped into a high-dimensional feature vector space by using the kernel function, and the linear inseparable classification problem of the original input space is transformed into the linear separable classification problem in the high-dimensional space, and the optimal classification hyperplaneis constructed in the high-dimensional space. Currently, SVRM has been applied in more and more fields due to its unique superiority [30,31].

The principles of SVRM can be described briefly. For nonlinear regression, a nonlinear function is used to map training data into a high-dimensional feature space and linear regression is carried out in the high-dimensional feature space. The regression estimation function is set as Formula (18), where ω represents the dimension of the feature space, ϕ(x) is the nonlinear function, *b* is the threshold. The optimization problem to be solved by SVRM could be described by Equation (19), where *c* is the penalty coefficient used to control the complexity of the model, training error rate and generalization ability, $\xi_{i},\xi_{i}^{*}$ are the relaxation factors, ε is the default error limit and *m* is the number of training samples:(18)f(x)=ω·ϕ(x)+b
(19)$$\underset{\omega ,b,\xi }{\min}\frac{1}{2}\left\|\omega \right\|+c\sum_{i=1}^{m}\left(\xi_{i}+\xi_{i}^{*}\right)$$
$$s.t.\{\begin{array}{l} \omega \cdot \phi (x_{i})+b-y_{i}\leq \xi_{i}^{*}+\varepsilon \\ \omega \cdot \phi (x_{i})+b-y_{i}\leq \xi_{i}+\varepsilon \;\;i=1,\;2,\;\ldots ,m\\ \xi_{i}^{*},\xi_{i}\geq 0,(i=1,2,\ldots ,m) \end{array}$$

The dual form of the optimization problem could be obtained as Equation (20):(20)$$\max L(a,a^{*})=-\frac{1}{2}\sum_{i=1}^{m}\sum_{j=1}^{m}(a_{i}-a_{i}^{*})(a_{j}-a_{j}^{*})K(x_{i}\cdot x_{j})+\sum_{i=1}^{m}y_{i}(a_{i}-a_{i}^{*})-\varepsilon \sum_{i=1}^{m}\left(a_{i}+a_{i}^{*}\right)$$
$$s.t.\sum_{i=1}^{m}(a_{i}-a_{i}^{*})=0\;0\leq a_{i},a_{i}^{*}\leq C,\;i=1,\;2,\;\ldots ,m$$
where *a_i_* and *a_i_^*^* are range multiplier, $K(x_{i}\cdot x_{j})=\phi (x_{i})\cdot \phi (x_{j})$ is the kernel function. The regression estimation function could be expressed as Equation (21):(21)$$f(x)=\sum_{i=1}^{m}(a_{i}-a_{i}{}^{*})K(x,xi)+b$$

Generally speaking, the most commonly used kernel types in SVRM includes linear, polynomial, radial basis function (RBF), sigmoid and pre-computed kernels. Since only one parameter needs to be determined in RBF and it has global performance similar to other kernel functions, RBF is chosen as the kernel function of SVRM here.

### 4.3. PSO-SVRM Model

There are two parameters to be optimized when using RBF as the kernel function, one is the penalty factor *C,* the other is the kernel width parameter σ. Both of these parameters have a great impact on the performance of SVRM. The penalty factor *C* controls the complexity of the SVRM model and the penalty level for the wrong samples. If *C* is too large, the prediction accuracy of SVRM in training phase is very high, but in testing phase is very low; if *C* is too small, it is difficult to obtain satisfactory prediction accuracy. The parameter σ reflects the complexity of the distribution of data samples in the high-dimensional feature space, which determines the complexity of the linear classification surface. It has a greater impact on the prediction accuracy of SVRM than the penalty factor *C*, too large value will lead to an “over-learning” problem, and too small value will lead to an “under-learning” problem. Therefore, in the SVRM training process, it is particularly important to choose the appropriate parameters for the penalty factor *C* and kernel function width σ. At present, PSO algorithm has been used to optimize the parameters of least squares support vector machines in order to construct an optimal LS-SVM classifies [32] and an improved adaptive particle swarm optimization algorithm has been proposed [33]. In this paper, PSO algorithm is used to automatically select the appropriate parameters of SVRM model and PSO-SVRM model is proposed to obtain the best performance.

The process of optimization of SVRM parameters based on PSO algorithm is shown as Figure 4. As could be seen from Figure 4, at the very beginning of the algorithm, the search range for SVRM model parameters is set. After that, the values of the population size, learning factors (*c_1_*, *c_2_*), weight factor (ω), the maximum number of iterations and the maximum speed value in the PSO algorithm are initialized. Then particle velocity and its position are initialized randomly. The fitness of each particle is calculated and the best individual and global particle position is updated. The velocity and the position of the particle is updated subsequently. Determine whether the given maximum number of iterations have been reached or the accuracy requirements have been achieved, if so, stop optimization, return the current optimal SVRM model parameters, otherwise go on with the iteration. A step by step description of the algorithm is shown as [Sec secA2-entropy-21-00117].

## 5. Results and Discussions

The PSO-SVRM model for fatigue life prediction of titanium alloy welded joints was established by using the LibSVM toolbox [34]. We set the local search capabilities related parameter *c*_1_ = 1.5, the global search capability related parameters *c*_2_ = 1.7, the maximum number of iterations of the algorithm was 2000 times, the maximum population of the particles was 20, the value of ω decreased linearly from 0.9 to 0.4 as the number of iterations increased. The PSO algorithm was used for optimization of the SVRM parameters *C* and σ. The SVRM model was established based on the best parameter values of *C* and σ to predict the fatigue life of the titanium alloy welded joints.

Since there is only one fatigue sample in the fatigue characteristic domain1, it is not discussed here. The PSO-SVRM model for fatigue life prediction of titanium alloy welded joints was established in the determined fatigue characteristic domains, from domain2 to domain5, respectively. The fatigue life prediction values of the titanium alloy welded joints by using the PSO-SVRM model were compared with that obtained by using the tradition least square fitting method and the actual fatigue life obtained through fatigue test. The comparison results are shown in Figure 5. The error of fatigue life prediction by using PSO-SVRM model and by using the traditional least square fitting method is shown in Figure 6.

As could be seen from Figure 5 that the blue ‘○’ indicates the actual fatigue life cycles in the fatigue test. The red ‘□’ denotes the fatigue life prediction value by using the PSO-SVRM model. The black ‘*’ indicates the fatigue life prediction value by using the least square method. Compared with the least squares method, the prediction results obtained by using the PSO-SVRM model are much closer to the actual fatigue life cycles.

As is shown in Figure 5 the black ‘*’ indicates the fatigue life prediction error of the least squares method and the red ‘□’ denotes the fatigue life prediction error of the PSO-SVRM model. It could be seen from Figure 5 that the fatigue life prediction error by using PSO-SVRM is less than using the least squares method. The fatigue life prediction results shown in Figure 5 and Figure 6 indicate that when PSO-SVRM model is used in the determined joint type fatigue characteristic domain, smaller prediction errors could be achieved than by using the tradition least squares method.

## 6. Conclusions

Entropy-based neighborhood rough set theory is used for analysis of the fatigue life influencing factors of titanium alloy welded joints and the PSO-SVRM model is constructed for fatigue life prediction of the titanium alloy welded joints. The following conclusions can be proposed:(1)The reduction results show that besides the equivalent structural stress range, the joint type influencing factors also play a very important role in determining the fatigue life of titanium alloy welded joints.(2)The fatigue characteristic domain could be determined according to the reduction results of the entropy-based neighborhood rough set theory.(3)A PSO-SVRM model for fatigue life prediction of titanium alloy welded joints is established in the determined fatigue characteristic domain. Experimental results indicate that compared with the traditional least squares method, the proposed PSO-SVRM model could better predict the fatigue life of the titanium alloy welded joints.

Future work will be concentrated on other determination methods of fatigue characteristic domain and further validation of the PSO-SVRM model as more fatigue test samples become available.

## Figures and Tables

**Figure 1 entropy-21-00117-f001:**
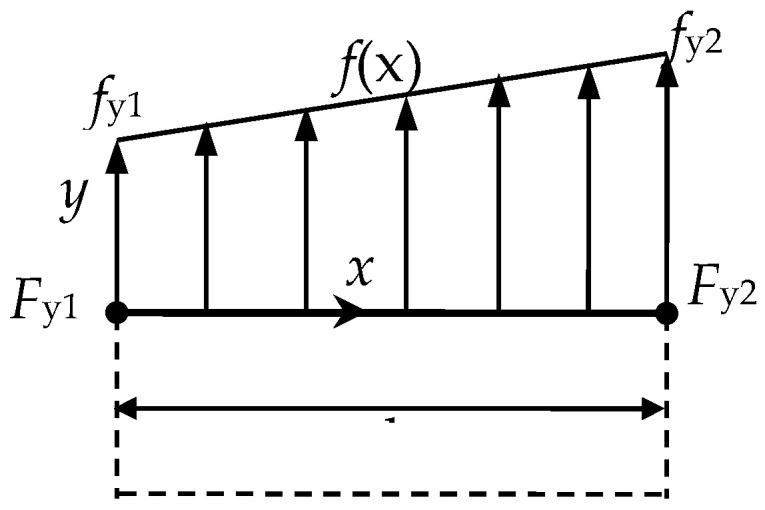
Definition of line force.

**Figure 2 entropy-21-00117-f002:**
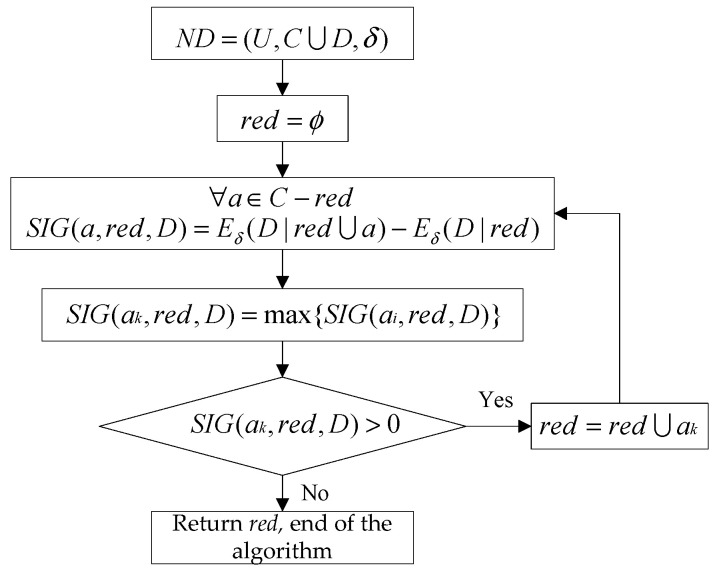
Neighborhood entropy-based reduction algorithm.

**Figure 3 entropy-21-00117-f003:**
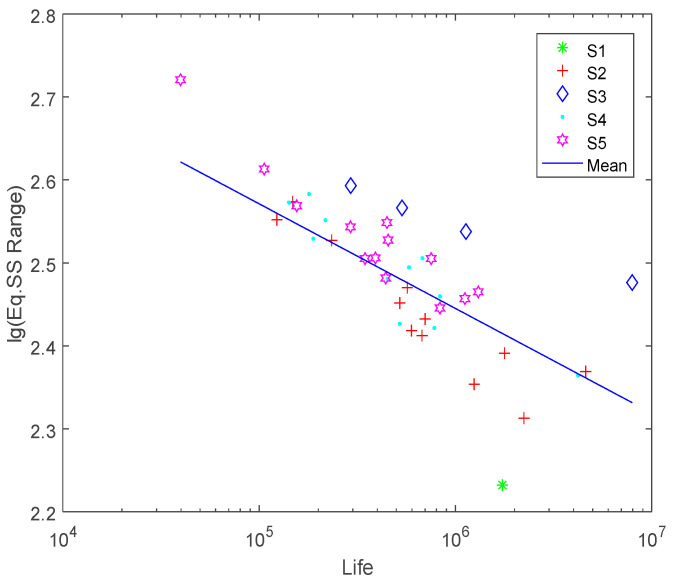
Characteristic domain of joint type.

**Figure 4 entropy-21-00117-f004:**
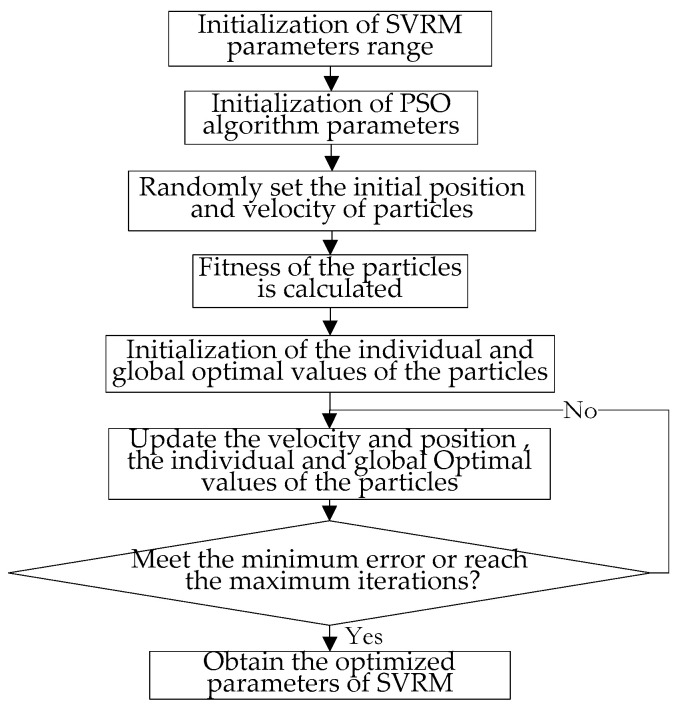
Optimization of SVM parameters based on the PSO algorithm.

**Figure 5 entropy-21-00117-f005:**
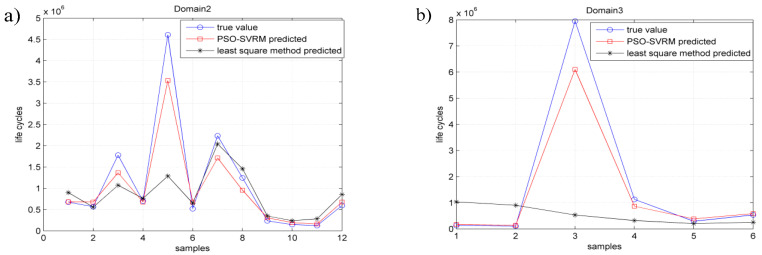
Life prediction results; (**a**) domain2; (**b**) domain3; (**c**) domain4; (**d**) domain5.

**Figure 6 entropy-21-00117-f006:**
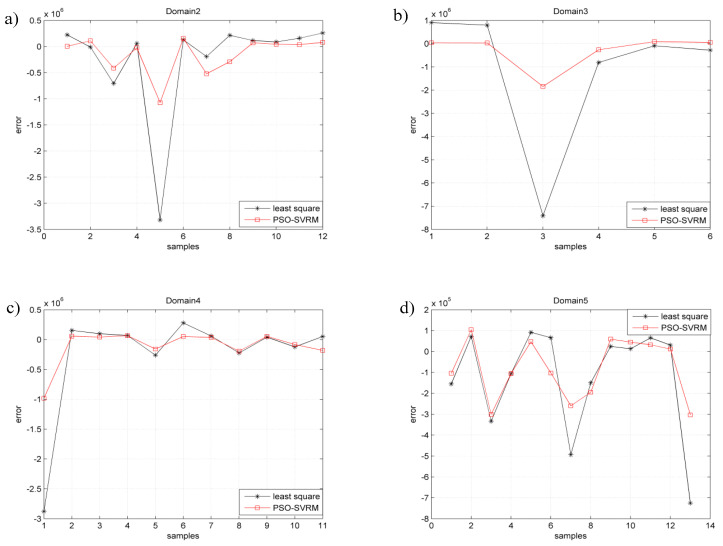
Comparison of the fatigue life prediction errors: (**a**) domain2; (**b**) domain3; (**c**) domain4; (**d**) domain5.

**Table 1 entropy-21-00117-t001:** Decision system of titanium alloy welded joints.

*U*	*J*	*t* (mm)	*r*	*M*	$\Delta S_{s}\;(\mathrm{MPa})$	*N* (Cycles)
*x* _1_	LB	2	0	JIS H4600	170.61	1734430
*x* _2_	LT	2	0	JIS H4600	258.4	675006
*x* _3_	CB	2	0	JIS H4600	248.51	129860
*x* _4_	LL	2	0	JIS H4600	353.73	447552
*x* _5_	LT	10	0	JIS H4600	205.54	2233310
*x* _6_	CB	10	0	JIS H4600	299.43	7946640
*x* _7_	CT	10	0	JIS H4600	288.15	833690
*x* _8_	LL	10	0	JIS H4600	278.98	833690
......						

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
