# Peer review of "An Entropy-Based Neighborhood Rough Set and PSO-SVRM Model for Fatigue Life Prediction of Titanium Alloy Welded Joints"

_entropy, 2019, doi:10.3390/e21020117_

Round 1

Reviewer 1 Report

Authors use information entropy neighborhood rough set and PSO-SVRM model for fatigue life prediction of titanium alloy welded joints. There is no literature survey in the paper. The manuscript gives the impression that this is the first time, entropy has been used for fatigue life prediction. This is not acceptable.  Entropy has been used in the past for fatigue life prediction since 1997. While I agree that entropy authors use is not thermodynamics entropy, it is still the same idea and concept.  I suggest authors review the papers listed below and explain how they differ from earlier work in their originality.

1-              Basaran, C., and Yan. C.Y."A Thermodynamic Framework for Damage Mechanics of Solder Joints,” Trans. of ASME, Journal of Electronic Packaging, Vol. 120, 379-384, 1998

2-              Basaran, C. and Nie, S.”An Irreversible Thermodynamic Theory for Damage Mechanics of Solids,” International Journal of Damage Mechanics, vol. 13, No 3, pp 205-224, July 2004,

3-              Leonid A. Sosnovskiy and Sergei S. Sherbakov, Mechanothermodynamic Entropy and Analysis of Damage State of Complex Systems, Entropy 2016, 18, 268; doi:10.3390/e18070268

4-               Mu-Hang Zhang , Xiao-Hong Shen , Lei He  and Ke-Shi Zhang, Application of Differential Entropy in Characterizing the Deformation Inhomogeneity and Life Prediction of Low-Cycle Fatigue of Metals, Materials, 2018, 11, 1917,, http://dx.doi.org/10.3390/ma11101917

5-              Basaran, C. and Tang, H.,”Implementation of a Thermodynamic Framework for Damage Mechanics of Solder Interconnects in Microelectronic Packaging,” International Journal of Damage Mechanics, Vol. 11, No. 1, pp. 87-108, January 2002.

6-              Gomez, J. and Basaran, C.”A Thermodynamics Based Damage Mechanics Constitutive Model for Low Cycle Fatigue Analysis of Microelectronics Solder Joints Incorporating Size Effect,” International Journal of Solids and Structures, Vol. 42, issue 13, pp. 3744-3772, (2005)

Author Response

Dear reviewer,

       Thanks a lot for the constructive comments .

       We have made supplements  to briefly review the research status of fatigue life prediction based on thermodynamics entropy.

       Another paragraph(from line 49 to line 70) has been added to the manuscript in the introduction section as" Thermodynamics entropy has been used for fatigue life prediction for a long time. For example, a damage metric based on the second law of thermodynamics and statistical mechanics was presented[8]. The proposed thermodynamic framework treated a solid body as a thermodynamic system and required that the entropy production be nonnegative. A thermodynamic framework was presented for damage mechanics of solid materials[9]. It was believed that there was no need for physically meaningless empirical parameters to define a phenomenological damage potential surface or a Weibull function to trace damage evolution in solid continuum. A generalized theory of evolution based on the concept of tribo-fatigue entropy was proposed where tribo-fatigue entropy was determined by the processes of damage ability conditioned by thermodynamic and mechanical effects causing to the change of states of any systems[10]. The Shannon’s differential entropies of both distributions of the strain in the loading direction and the first principal strain were employed at the tension peak of the cycles as measuring parameters of strain inhomogeneity[11]. A thermo mechanical fatigue life prediction model based on the theory of damage mechanics was presented[12]. The damage evolution, corresponding to the material degradation under cyclic thermo mechanical loading, was quantified thermodynamic framework. A viscoplastic constitutive model unified with a thermodynamics based damage evolution model was embedded into a couple stress framework in order to simulate low cycle fatigue response coupled to size effects[13]. Entropy, as a state function of thermodynamics, is independent of the failure path and can deal with the uncertainties related to the damage threshold[14] (such as the systemic chaos caused by performance degradation). Entropy generation and entropy accumulation are taken as the measurement of performance degradation [15], and the effectiveness of using constant irreversible entropy increment to evaluate metal fatigue damage [14] is also gradually verified."      

        While in this work, based on information entropy,  the concept of the fatigue characteristics domain is proposed and a PSO-SVRM model for fatigue life prediction of titanium alloy welded joints is suggested.

        Hope the revisions would meet with your approval. Thanks.

        Best wishes.

Reviewer 2 Report

The manuscript needs to be revised thoroughly before it is accepted in Entropy. In the current form, it appears like a technical report which is very difficult to comprehend. I would reccomend to give the steps in algorithms as supplimentary files rather than putting it as a part of manuscript. 

Author Response

Dear reviewer,

   Thanks a lot for your constructive comment.

   We have tried our best to revise the manuscript. On one hand, one more paragraph has been added to the paragraph in the introduction paragraph to briefly review the research status of fatigue life prediction based on thermodynamics entropy. On the other hand, all the steps in algorithms has been put in Appendix A. Please find it in the revised manuscript. Hope this revision would meet with your approval.

 Thanks a lot!

Reviewer 3 Report

This article deals with  titanium welded joints and reports their fatigue life prediction obtained by entropy based neighborhood rough set reduction algorithm, PSO (particle swarm optimization) algorithm and SVRM (support vector regression machine). Authors characterized in detail proposed mathematical models and calculations results in reference to the tradition least square fitting method and the actual fatigue life obtained through fatigue test. The manuscript contains interesting information but requires additional explanations before publishing:

1.       What was type of titanium used  - Gr1 or Gr2?

2.       Why was CP titanium analyzed? Two-phase titanium alloy (e.g. Ti6Al4V) are usually used as a construction material.

3.       Are the proposed mathematical models useful for two-phase and near-beta titanium alloys?

4.       Do mathematical models take into account initial state of joined material (e.g. rolling, hot rolling, hot forging, etc.)?

5.       How do mathematical models include macro/microstructural parameters (width, grain shape and size, phases composition, etc.) of Heat Affected Zone (HAZ) in welded joint?  

6.       Table 1 – please use MPa as a unit of equivalent structural stress

7.       Line 173, 174 – What does “equivalent structural stress range” mean?  What is min/max value of this range?

Author Response

Dear reviewer,

    Thank you very much for your constructive comments.

    Please find the point-by-point response in the attachment.

    Hope the response and revision would meet with your approval. 

    Best wishes.

Round 2

Reviewer 1 Report

Authors incorporated all my comments

Reviewer 3 Report

I suggest accepting article in present form.